# Accelerating DC Circuit Simulation through Feature Selection and LSTM-Based Time-Step Control

Zhou Jin [1], Ziyi Yang [1], Haojie Pei [1], Xiaru Zha [1], Yinuo Bai [1], Dan Niu [2], Zhenya Zhou [3] and Xiao Wu [3,*]

1   Super Scientific Software Laboratory, China University of Petroleum-Beijing, Beijing 102249, China; jinzhou@cup.edu.cn (Z.J.); 2020011629@student.cup.edu.cn (Z.Y.); 2020215948@student.cup.edu.cn (H.P.); 2019011712@student.cup.edu.cn (X.Z.); bai@student.cup.edu.cn (Y.B.)
2   School of Automation, Southeast University, Nanjing 211189, China; 101011786@seu.edu.cn
3   Huada Empyrean Software Co., Ltd., Beijing 100102, China; zhouzhy@mail.empyrean.com.cn
*   Correspondence: wuxiao@mail.empyrean.com.cn

**Abstract:** Circuit simulation has become increasingly significant in circuit design with the development of very large scale integration, and direct current (DC) analysis, which serves as the basis of circuit behavior analysis, is the foundation for nonlinear electronic circuit simulation. Among the several continuation algorithms for DC analysis, pseudo-transient analysis (PTA) methods have gained great success. However, PTA tends to be computationally intensive without a proper time-step control method. In order to improve this problem, we propose a novel time-step control method enhanced by advanced deep learning in this paper. Specifically, a coarse and fine-grained hybrid sampling strategy is introduced to find the optimal time step, which resolves the problem that the optimal time step has no precise definition in PTA theory. After that, a long short-term memory (LSTM) network, with the ability to process temporal information, can be employed to learn the optimal time-step control method based on feature selection and a two-stage data preprocessing strategy, which accelerates DC analysis. Furthermore, random forest (RF) is also used to evaluate feature importance, which can achieve feature selection with reduced dimensions, thereby speeding up the network's training speed and improving the accuracy of prediction. Experimental results demonstrate a significant speedup: up to 61.32 times.

**Keywords:** DC analysis; pseudo-transient analysis; time-step control; deep learning

## 1. Introduction

The progress of semiconductor and computer technology has resulted in the development of highly integrated and complex integrated circuits, exhibiting exponential growth. To simulate these circuits, DC analysis based on modified nodal analysis (MNA) [1] is essential for solving nonlinear algebraic equations. However, the convergence of the widely-used Newton–Raphson (NR) method [2] and its variants [3,4] depends heavily on the diagonal dominance of the coefficient matrix and the appropriateness of the initial point. These factors are difficult to guarantee for analog circuits, often leading to convergence failure [5]. As a result, researchers have investigated continuation algorithms to address the NR convergence problem, including Gmin stepping [6], source stepping [7], homotopy methods [8–10], and PTA [11]. However, Gmin and source-stepping methods may have inferior convergence when the solution curve is bifurcated, folded, or discontinuous; and the practical implementations of homotopy methods are highly dependent on device models. PTA has shown to be a promising alternative due to the good continuity of its solution curve and easy implementation.

PTA is used to convert a complex nonlinear algebraic system into an ordinary differential system, which can be solved with initial value problems by inserting pseudo-elements. Once the PTA solver has formed the ordinary differential system, it is solved iteratively through numerical integration using a time-step control technique to achieve a steady state.

Since the efficiency of PTA is heavily influenced by its time-step control method, which determines the discrete time points that require resolution, including the resource-intensive NR iterations, it is crucial to optimize this aspect. Although some simplistic formula-based time-step control methods have been introduced in recent years to speed up PTA, such methods have not been adequate for large-scale nonlinear simulations. Therefore, a more effective time-step control approach is required.

The emergence of deep learning technologies has enabled the resolution of complex problems, such as computer vision [12], cloud computing [13], and natural language processing [14], offering a promising avenue for researching time-step control methods. However, developing an efficient time-step control approach presents several challenges. Firstly, while different circuit types require different time-step requirements, conventional process variables in circuit solving can be utilized as features, along with expert knowledge, to identify the distinct time-step necessities. Secondly, since there is no precise definition for the optimal time step in PTA theory, a sampling strategy is necessary to determine the optimal time step. Thirdly, because PTA time-step control is temporal in nature, the proposed algorithm must be able to process timing information.

In this paper, we present a novel approach to time-step control by leveraging deep learning techniques to address Challenge 1. Our proposed method is illustrated in Figure 1 and features a hybrid searching strategy with coarse and fine-grained steps to find the optimal time-step, thereby addressing Challenge 2. In addition, we utilized the LSTM model to effectively manage timing information and resolve Challenge 3. Our contributions are summarized as follows:

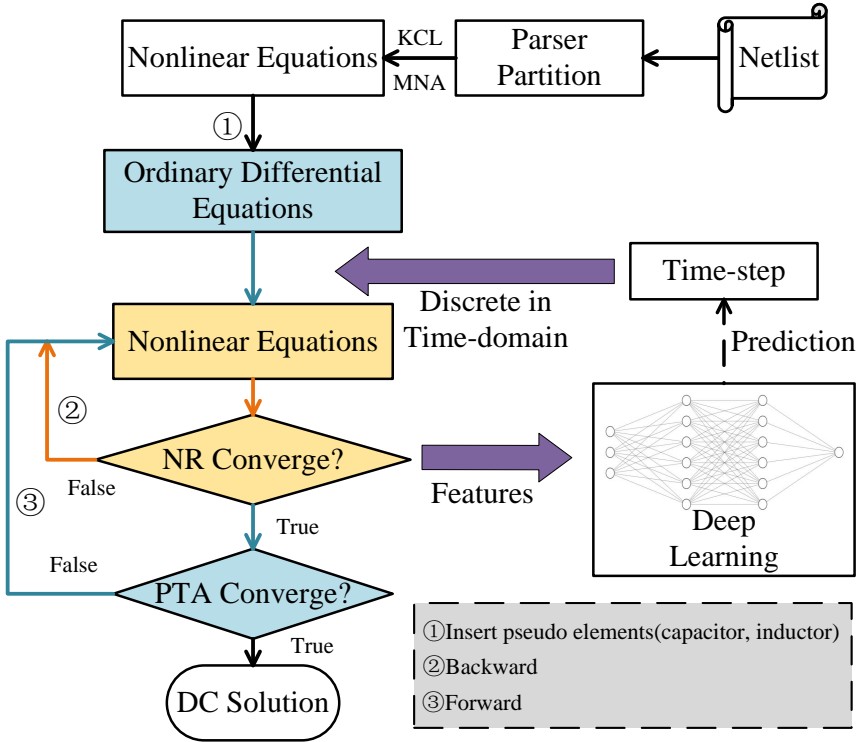

**Figure 1.** The framework of the time-step control method enhanced by deep learning.

(1) We present a novel time-step control method that incorporates a deep learning framework for offline training and online prediction. Our methodology yields superior simulation efficiency and exhibits remarkable out-of-sample performance, surpassing extant approaches in terms of extrapolation accuracy.

(2) We propose a hybrid searching strategy for identifying the optimal time-step, addressing the key issue that cannot be theoretically defined. This approach also provides fine-grained labels for the training set.

(3) We incorporate the random forest model to evaluate feature importance during the feature-selection stage, leading to faster network training, improved simulation efficiency, and highly accurate predictions.

(4) We implemented our proposed approach in an out-of-the-box SPICE-like simulator and demonstrate its effectiveness with benchmark circuits. We achieved significant acceleration—the maximum speedup being 61.32 times on practical circuits.

## 2. Preliminaries

### 2.1. Pseudo-Transient Analysis

Conventional PTA methods mainly include pure PTA [15,16], damped pseudo-transient analysis (DPTA) [17], and compound-element pseudo-transient analysis (CEPTA) [18,19]. Pure PTA methods insert pure capacitors or inductors into a circuit as pseudo-elements, whose values are either constant or time-varying. With appropriate insertion element values and time-step control algorithms, pure PTA methods can effectively solve the discontinuous problem. However, pure elements may cause oscillation problems. Among these three methods, DPTA uses a damped numerical integration method to artificially increase the damping effect, thereby greatly reducing oscillation. Compared with the pure PTA method, CEPTA uses compound branches as pseudo-elements, which have also been proved to be effective at eliminating oscillation. Nonetheless, waveform continuity cannot be guaranteed with CEPTA.

From the circuit equation viewpoint, PTA inserts specific pseudo-elements, including capacitors and inductors, into the original circuit, transforming the original hard-to-solve nonlinear algebraic system:

$$\boldsymbol{F}(\boldsymbol{x}) = 0 \tag{1}$$

($\boldsymbol{F}(\cdot) : \mathbb{R}^m \to \mathbb{R}^m$, $\boldsymbol{x} = (\boldsymbol{v}, \boldsymbol{i})^T \in \mathbb{R}^m$, $m = N + M$, $\boldsymbol{x}$ represents the vector to be solved, variable vector $\boldsymbol{v} \in \mathbb{R}^N$ denotes the node voltages, and vector $\boldsymbol{i} \in \mathbb{R}^M$ represents internal branch currents) into the ordinary differential system

$$\boldsymbol{F}(\boldsymbol{x}) + \boldsymbol{D} * \dot{\boldsymbol{x}}(t) = 0 \tag{2}$$

(where $\dot{\boldsymbol{x}}(t) = (\dot{\boldsymbol{v}}(t), \dot{\boldsymbol{i}}(t))$, $\dot{\boldsymbol{x}}$ denotes the differential of node voltage and internal current with time and $\boldsymbol{D}$ represents the incidence matrix of inserted pseudo-elements) with an initial value problem.

Use implicit numerical integration algorithms, which are shown in Equation (3), to discrete Equation (2) in the time-domain and finally get the steady state through difference approximation of the differential term iteratively.

$$\dot{\boldsymbol{x}}(t)|_{t=t_{n+1}} = (\boldsymbol{x}_{n+1} - \boldsymbol{x}_n)/h_{n+1} \tag{3}$$

How to select time-step size $h$ (as shown in Equation (3)) for each PTA iteration is known as the time-step control method. In addition, Equation (3) can support the conclusion mentioned above: the time-step control method in PTA determines the number of equations that need to be solved at discrete time points, which involves time- and resource-consuming NR iterations. Generally, an efficient time-step control method can not only reduce the number of NR iterations but also decrease the number of PTA rejected steps due to non-convergence of NR iterations.

### 2.2. Time-Step Control Method

There are two time-step control methods based on simple formulas. Conventional PTA methods use a simple iteration counting method [17] to determine time-step size. This method implements time-step control through two options (IMAX and IMIN). It compares the number of NR iterations at each time point with options to determine the next time-step. The advantage of this method is that the time-step can be increased simply and quickly. However, it is a very difficult problem to select appropriate parameters for different circuits, including IMAX, IMIN, initial time-step, time-step growth rate, etc.

Another adaptive time-step control method based on switched evolution/relaxation (SER) was proposed in [20], which employs Equation (4) to control the time step.

$$h_{n+1} = E(h_n, Nitr_n, \mathbf{x}, F(\mathbf{x}))$$
$$= h_n \cdot MAX(1, \delta \cdot \gamma \cdot \|F(\mathbf{x}_{n-1})\| / \|F(\mathbf{x}_n)\|) \tag{4}$$

where $\delta$ determines the relative change of $\mathbf{x}$ per time step, $\gamma$ assesses the difficulty level for the convergence of NR in the previous steps, and $\|F(\mathbf{x}_{n-1})\| / \|F(\mathbf{x}_n)\|$ represents the residuals' reduction. It is a heuristic method that employs domain experiences, demonstrating great potential in speedup that can be obtained through intelligent time-step control. However, the heuristic method still cannot guarantee that the time-step will always be as large as possible.

### 2.3. Random Forest

Random forest [21] is an ensemble learning algorithm that belongs to the bagging type. By combining multiple weak classifiers, the final results are voted or averaged, which makes the results of the whole model have high accuracy and generalization performance. The success of this method can be attributed to the use of both "random" and "forest". The former helps to prevent overfitting, and the latter improves accuracy. Figure 2 shows the training process of the random forest.

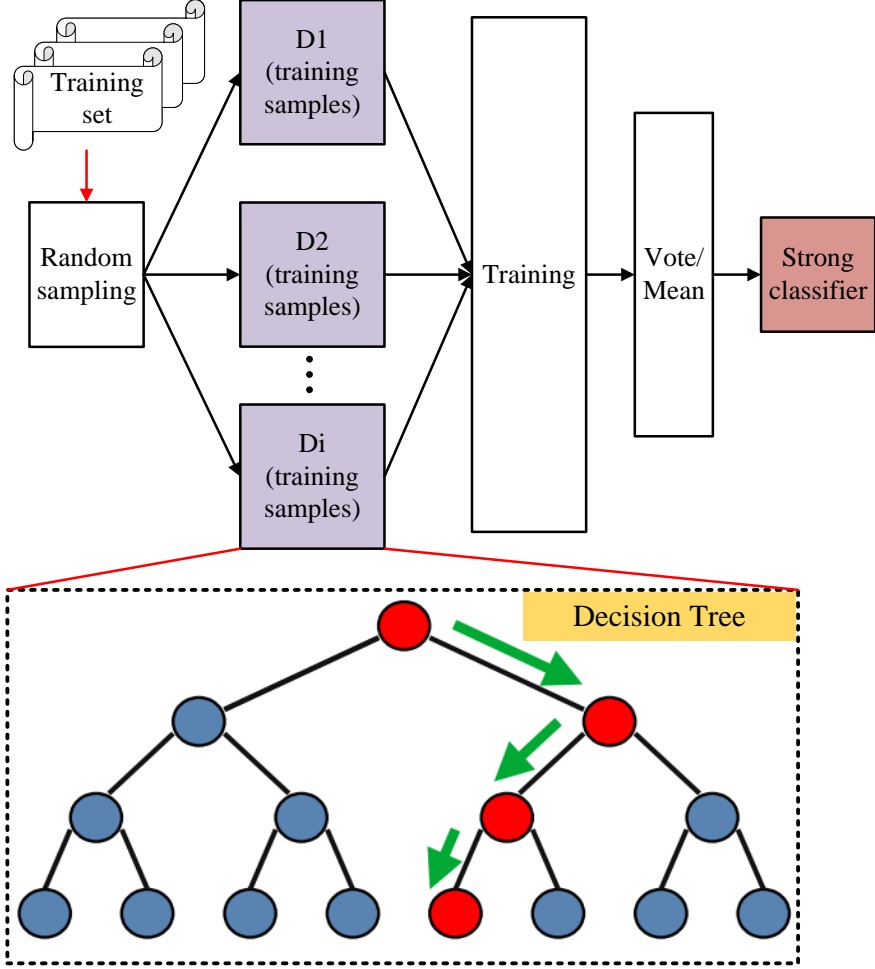

**Figure 2.** The training process of the random forest.

### 2.4. Long Short-Term Memory Network

LSTM, as shown in Figure 3, is a special recursive neural network [22,23] which can deal with the problem of gradient disappearance in time-sequence information from back propagation and successfully be applied to complex computation and classification tasks in various fields [24]. It is widely used for sequential tasks, such as speech recognition and machine translation. LSTM consists of three gate structures, including a forget gate, an input gate, and an output gate, which realize the protection and control of timing information. The forget gate shown in Equation (5) is used to determine what information needs to be discarded.

$$f_t = \sigma(W_f \cdot [h_{t-1}, x_t] + b_f) \tag{5}$$

where $f_t$ indicates the output of the forget gate, $W_f$ and $b_f$ are the weight and bias of the forget-gate neuron, and $\sigma$ represents the sigmod activation function. The input gate described in Equation (6) is used to determine what new information needs to be retained and what information needs to be updated.

$$
\begin{aligned}
i_t &= \sigma(W_i \cdot [h_{t-1}, x_t] + b_i) \\
C'_t &= tanh(W_C \cdot [h_{t-1}, x_t] + b_C) \\
C_t &= i_t \cdot C'_t
\end{aligned}
\tag{6}
$$

where $i_t$ indicates the partial input of the input gate through the sigmod function; $C'_t$ represents the partial input of the input gate through the tanh function; $C_t$ shows final input of the input gate; and $W_i$, $b_i$, $W_c$, and $b_c$ represent weight and bias of the input-gate neuron.

The output gate represented in Equation (7) is used to determine which information needs to be passed to the next cell.

$$
\begin{aligned}
o_t &= \sigma(W_o \cdot [h_{t-1}, x_t] + b_o) \\
h_t &= o_t \cdot tanh(C_t)
\end{aligned}
\tag{7}
$$

where $o_t$ indicates the partial output of the output gate, $h_t$ represents the output for the next LSTM cell, and $W_o$ and $b_o$ are the weight and bias of the output-gate neuron. In all of the above equations, $h_{t-1}$ represents the output of the previous layer, and $x_t$ represents the input of the current cell.

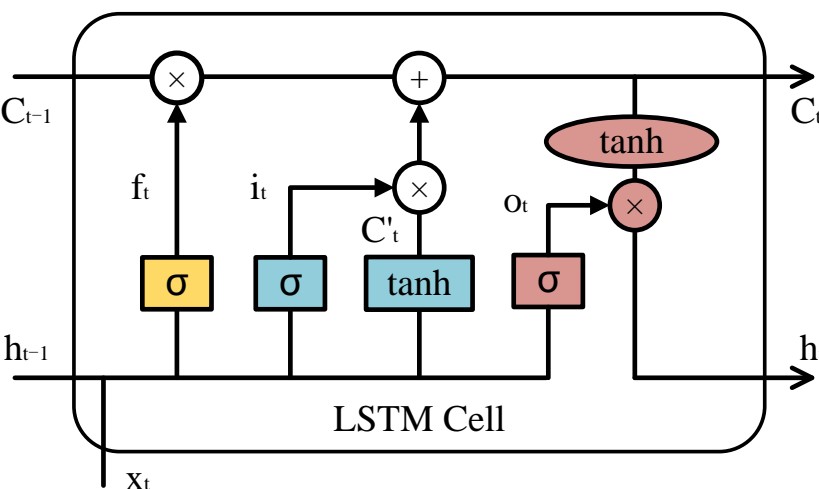

**Figure 3.** The construction of the LSTM cell.

## 3. Proposed Methods

### 3.1. Overview

The time-step control for the PTA method is not determined by accuracy considerations. Instead, the time-step is made as large as possible, which is consistent only with the convergence of the NR iteration [20], and this time-step is defined as the optimal time step. In the actual simulation, if each PTA step can use the optimal time-step, the simulation efficiency can be greatly improved. According to the characteristics of the PTA time-step mentioned above, we first introduce a hybrid searching strategy for finding the optimal time step. Specifically, the hybrid search strategy consists of two parts: the coarse-grained process and the fine-grained process. The coarse-grained process is responsible for increasing the time step rapidly and without limitation under the precondition of NR convergence. The fine-grained process is triggered when NR does not converge and is responsible for reducing the time step, leading to NR non-convergence from large to small according to the set granularity, until a time step is found that can ensure NR convergence. In this case, the optimal time step is artificially defined—that is, the data labeling is completed. Furthermore, feature selection based on random forest is adopted to obtain fine features, which not only reduces the feature dimension through evaluating feature importance but also improves simulation efficiency with high accuracy of prediction and accelerates the network training.

Once optimal time steps and fine features are obtained, we utilize the key concepts of the proposed method—first, mapping time-step control for the regression-prediction problem. Then, the proposed method can fit the optimal time-step control function ($f$) with the optimal time step ($h$) and selected features ($s$) on the training set.

$$h = f(\boldsymbol{s}, \theta) \tag{8}$$

where $\theta$ represents the parameters needed to learn by training. Furthermore, LSTM is employed to find parameters $\theta^*$ on the training set, which makes model $f(\boldsymbol{s}, \theta^*)$ have a closed actual optimal time-step control function $f(\boldsymbol{s}, \theta)$ through batch gradient descent. Thus, the ordinary differential system reaches the steady state sufficiently quickly; that is, the number of NR iterations used to complete PTA is as small as possible.

Figure 4 shows the entire flow of the proposed method. Samples were collected during the PTA iteration for offline model training, and then the trained model was used for online prediction during the PTA iteration.

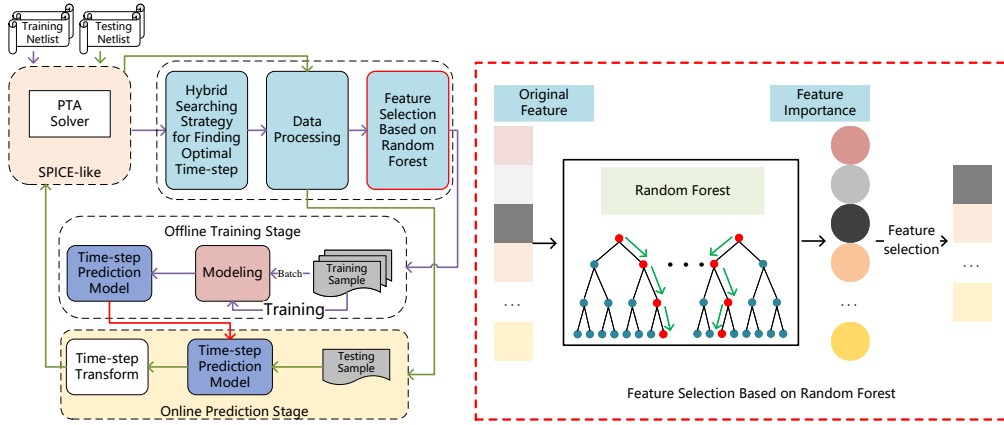

**Figure 4.** Entire flow of the proposed method.

### 3.2. Hybrid Searching Strategy for Finding the Optimal Time Step

As mentioned previously, searching for the optimal time step is the first task to be solved. However, there is the issue that there is no precise definition of the optimal time step in PTA. Therefore, we introduce a hybrid searching strategy, as shown in Figure 5, for

finding the optimal time step to overcome above issue. In the traditional PTA iteration, when solving the NR iteration convergence of the nonlinear equation at each time point, the solution of this time point will be received and taken as the initial value to continue solving the problem at the next time point until PTA convergence. In our hybrid search strategy, when we start from a certain point in time to find the optimal time step of the next step, we first use the traditional method to predict the next time step and solve it. When the time point solution of the predicted time step is solved, NR iteration converges and we do not accept the solution of this time point. Instead, it goes back to the time point where the optimal time step is to be found, and a larger time step is given for solving. This process is repeated until the NR iteration does not converge. When the NR solutions do not converge, we believe that the optimal time step must be between the non-convergent time point and the previous convergent time point. However, if the time step of NR iteration convergence in the previous step is taken as the optimal time step directly, it is not accurate and not the real optimal time step. At this time, we further use the fine-grained time step to search between these two time points, and the time step obtained can at least ensure that the convergence time is longer than the previous time step. To ensure the efficiency of searching strategy, an adaptive granularity trick for a fine grained process is adopted, which can choose different granularities according to the order of magnitude of time step. In this way, we can find the optimal time step at each time point by mixing the aggressive coarse-grained growth strategy while looking for a larger time step with the finer fine-grained rollback strategy, and obtain a large number of samples with the optimal time-step label through mass sampling, which makes subsequent supervised learning based on deep learning possible.

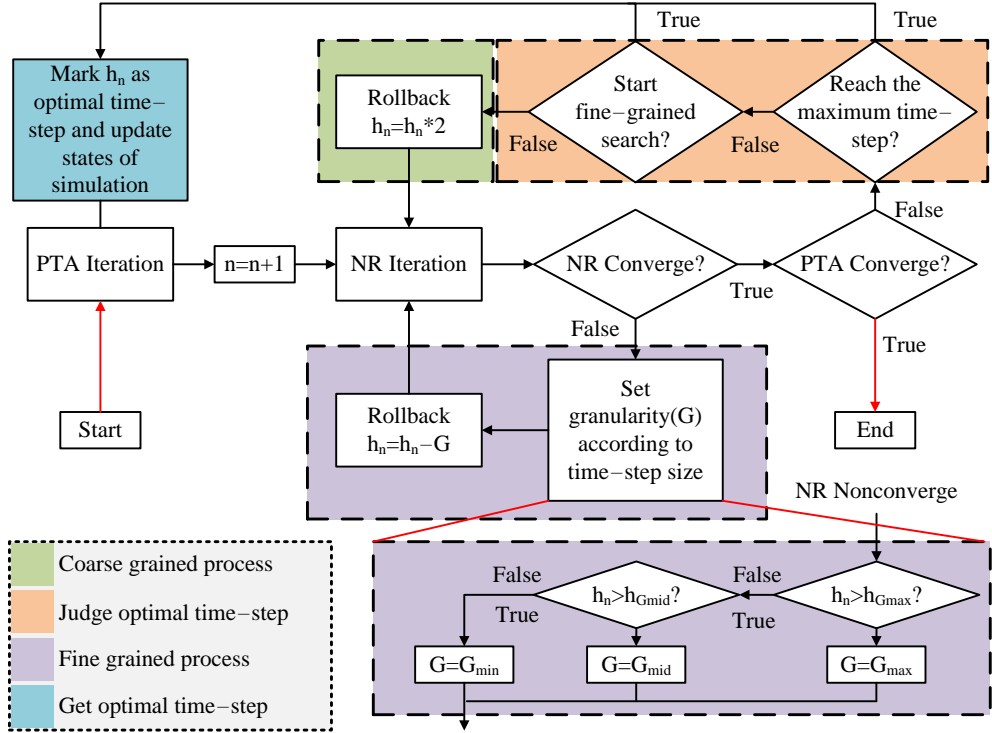

**Figure 5.** Hybrid searching strategy for finding the optimal time step. The symbol *n* indicates PTA iteration steps; $h_{Gmid}$ and $h_{Gmax}$ are threshold values for judging granularity(G); and *Gmid*, *Gmin*, and *Gmax* represent different granularity for sampling efficiency.

Table 1 shows the speedup of several circuits simulated by using the introduced sampling strategy, which verifies that this sampling strategy can provide a fine dataset for implementing the optimal time-step control enhanced by deep learning.

**Table 1.** Verifying the validity of the proposed sampling strategy on several circuits.

| Circuit | Conventional | Proposed Sampling | Speedup |
|---------|-------------|-------------------|---------|
| fadd32 | 1968 | 121 | 16.26 |
| ab_opamp | 2417 | 213 | 11.35 |
| ab_integ | 4540 | 159 | 28.55 |
| schmitfast | 5681 | 68 | 83.54 |
| THM5 | 5331 | 80 | 66.64 |

*3.3. Feature Selection Based on Random Forest*

Feature selection decides the performance of the trained model. Reasonable feature selection not only reduces the computation of model training but also improves the accuracy of prediction. In addition, once feature selection is completed, we simulate all training netlists to get the total dataset.

The time-step control in PTA is not limited to the circuit but depends on the change trends of the process variables in the simulation. Therefore, not only the features from the circuit itself, such as circuit type, but also the process variables in simulation, are selected as features. This enables our sample set not to be limited to a certain circuit type but can be sampled in the simulation process of all circuits in order to get as many samples as possible. Note that the features from five time points successively predict the sixth time step because too-few time points cannot describe the voltage fluctuation of nodes well, and in order to unify the inconsistent numbers of features caused by the different numbers of nodes in different circuits, we uniformly select the ten solution curves with the largest fluctuation for each circuit.

In our task, based on PTA theory and expert experience, we first select 17 features that are related to the time step (including step number; multiples of the time step; residuals of equations and NR iterations at the previous time point, and the standard deviation and mean of the residual; the 10 solution curves with the largest standard deviation; and the mean of the second-order norm of every solution curve along the row direction in a sliding window). Furthermore, random forest is employed to evaluate the importance of each feature. Each decision tree in a random forest is independent of the other. The information divergence is adopted to implement splitting of nodes in a decision tree, which shows the difference between the entropy of the set to be classified and the conditional entropy of the selected feature. Specifically, a random forest consists of many decision trees, which randomly select samples for training. The result of the random forest as a predictor (or classifier) is the mean value (mode) of each decision tree's output. Therefore, it has great advantages over other traditional machine learning algorithms as follows: (1) It can handle higher-dimensional data without feature reduction. (2) Unbiased estimation is used for the generalization error, and the model has a strong generalization ability. (3) The training speed is fast, and it is easy to parallelize. It can be clearly seen that Figure 6 shows a score of feature importance based on random forest to the 17 features mentioned earlier. Finally, the seven features with the higher scores, as shown in Table 2, were selected as inputs to train the LSTM network. Although the number of NR iterations was slightly lower than the values of other features (proportionally), it was still chosen as the training feature, because the number of NR iterations is the only feature that can represent the degree of difficulty of solving the equations. Our training sets were composed of 720 samples from 30 circuits to analyze the correlation between each feature and the label, so as to obtain the importance of each feature in the time-step prediction task. In addition, these circuits included a variety of different types of circuits, such as MOS, BJT, oscillating circuits, and convergence difficulty circuits.

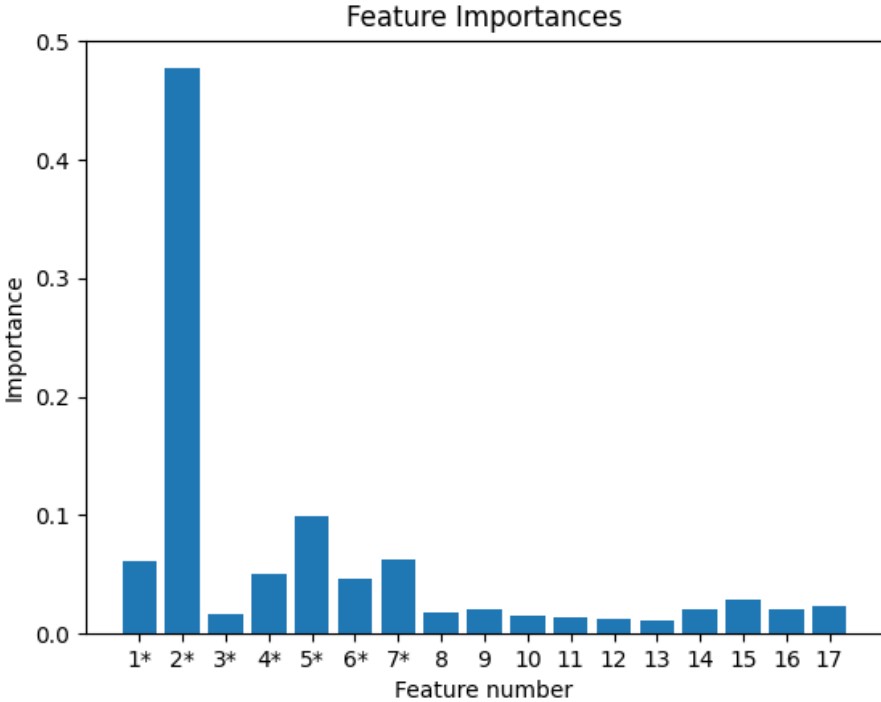

**Figure 6.** Feature importance according to a random forest. The descriptions of features 1* to 7* are shown in Table 2. The feature numbers 8 to 17 represent the standard deviations of the 10 solution curves with the greatest fluctuations.

**Table 2.** The features selected using the random forest.

| Feature Number | Feature Name | Description | Data Structure |
|:---:|:---:|:---:|:---:|
| 1* | $step_{n-1}$ | Evaluate the completion degree of the simulation | scalar |
| 2* | mul | The multiple of previous optimal time-step | scalar |
| 3* | $NRs_{n-1}$ | The NR iterations of previous optimal time-step | scalar |
| 4* | $Res\_std_{n-5:n-1}$ | The standard deviation of residuals in sliding window | vector |
| 5* | $Res_{n-1}$ | The residual of previous optimal time-step | scalar |
| 6* | $Res\_mea_{n-5:n-1}$ | The mean of residuals in sliding window | vector |
| 7* | $Vol\_norm\_mean_{1:5,1:all}$ | The mean value of the $norm_2$ of all voltage solution curves in sliding window | matrix |

The detailed descriptions of the main features are given follows:

$NRs_{n-1}$ represents the difficulty of NR convergence at the previous optimal time-step. A smaller value for it means that NR converges more easily, and the next time step can be larger.

$Res_{n-1}$ represents the distance of PTA convergence. When the value of the residual enters the PTA-convergence stage, a larger time step can also guarantee NR convergence.

$Vol\_norm\_mean_{n-5:n-1,1:all}$ represents the mean value of the $norm_2$ of all voltage solution curves in the sliding window because there is a certain relationship between the fluctuation of voltage-solution curves and the time step. In general, the more dramatic the voltage fluctuation, the smaller the time-step.

### 3.4. Data Preprocessing

**Individual data preprocessing for each feature**. From the data structure in Table 2, we can clearly know that different features have different data structures. However, the model requires input features with a uniform data structure (one-dimensional row vectors). Therefore, for residuals and voltages, we need to be unified. Firstly, for the voltage with a matrix structure, in order to preserve as much information as possible about the solution curve for all nodes $Vol_i$, the second order norm $Vol^{norm2}$, as shown in Equation (9), is applied to the resulting voltage row vector at each time point in the sliding window. At

this time, the matrix is changed to a vector with size *k* of sliding window, and the mean value $Vol_{mean}$ is adopted to normalize a scalar.

$$Vol^{norm2} = \sqrt{\sum_{i=1}^{n} Vol_i^2}$$

$$Vol_{mean} = \frac{\sum_{j=1}^{k} Vol_j^{norm2}}{k} \tag{9}$$

Secondly, for residuals of one-dimensional column vector type in size *k* of a sliding window, in order to describe the fluctuation of convergence distance of the current equations, the standard deviation $Res_{std}$ shown in Equation (11) and mean value $Res_{mean}$ shown in Equation (10) are used to normalize the residuals and yield two scalars, respectively. After that, we can splice independent features based on column direction into a one-dimensional row vector as a training sample.

$$Res_{mean} = \bar{Res} = \frac{\sum_{i=1}^{k} Res_i}{k} \tag{10}$$

$$Res_{std} = \sqrt{Res^2} = \sqrt{\frac{\sum_{i=1}^{k} \left( \dot{Res_i} - \bar{Res} \right)^2}{k-1}} \tag{11}$$

**Whole data preprocessing for training set**. After individual data processing, we spliced all the processed one-dimensional row vectors into a large matrix that is the training set, as shown in Figure 7. As we all know, for the training set, the different value ranges and dimensions of each column feature will increase the training time and even lead to the non-convergence of the model. Therefore, for each column feature *x* of the training set, the maximum and minimum normalization, as shown in Equation (12), should be used to carry out numerical unification. $X_{min}$ indicates the minimum value of the current column, and $X_{max}$ indicates the maximum value of the current column.

$$x' = \frac{x - X_{min}}{X_{max} - X_{min}} \tag{12}$$

It should be noted that the range of time steps may differ by dozens of orders of magnitude, which undoubtedly increases the difficulty of model learning. Therefore, in our work, in order to simplify the learning difficulty for the model, the time-step prediction task is converted to a prediction task based on the multiples of the previous time step.

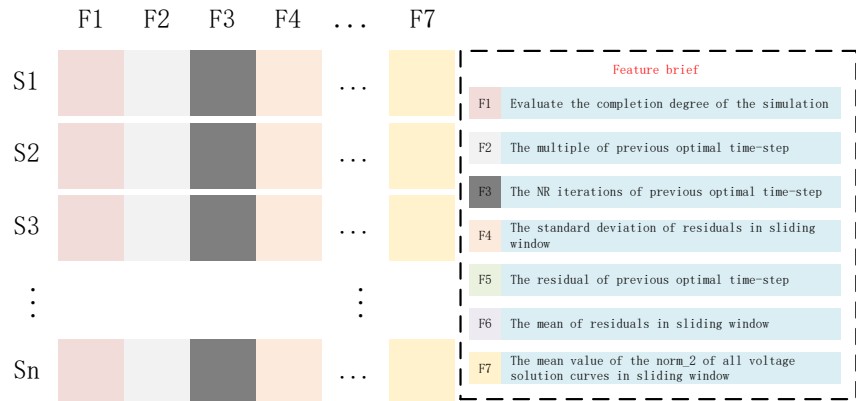

**Figure 7.** Diagram of the training set and feature brief.

### 3.5. LSTM-Enhanced Time-Step Control

Once all the data were ready to be sampled and processed, the deep learning model (LSTM) was employed to accomplish our task because of its ability to process timing information and avoid the gradient-disappearance problem in the recurrent neural network (RNN). PyTorch, a famous machine learning library, was adopted to implement LSTM. Firstly, we constructed a neural network structure with four hidden layers and a ReLU activation function, and each hidden layer contained 120 cells. In addition, batch size and learning rate were set to 32 and 0.0005, respectively. Adam and mean square error were utilized for the optimizer and loss function, respectively. Finally, Algorithm 1 was used for dealing with data and to train models. After training, the model for predicting time-step was obtained.

---

**Algorithm 1** LSTM-enhanced time-step control method for PTA.

---

**Input:** Training netlists $\xi$
**Output:** Time-step predictor $f(s, \theta^*)$
 1: **Hybrid Searching Strategy for Finding Optimal Time-Step**
 2: Construct nonlinear equation $F(x)$ by $\xi$
 3: **for** PTA is not converge **do**
 4:     Execute $NR_{solver}F(x)$
 5:     Find the maximum time-step that ensures NR convergence and mark as the optimal time-step
 6: **end for**
 7: Obtain optimal time-step set $H_{1:n}$ and features $S_{1:n,1:k}$
 8: **Feature Selection Based on Random Forest**
 9: Select features $S\_sub_{1:n,1:m}$, and ensure $m < k$
10: **Data Preprocessing**
11: Execute $S'\_sub_{1:n,1:m}, \hat{H}_{1:n} = Individual(S\_sub_{1:n,1:m}, H_{1:n})$
12: Execute $\hat{S}\_sub_{1:n,1:m} = Whole(S'\_sub_{1:n,1:m})$
13: **Modeling and Training**
14: Construct LSTM model with trainable parameters $\theta$
15: **for** i to n / $Batch$ **do**
16:     Loss (LSTM ($\hat{S}\_sub_{1:m,i:i+Batch}, \hat{H}_{1,i:i+Bactch}, \theta)$)
17:     Update $\theta \leftarrow \theta$
18:     Update $i \leftarrow i + Batch + 1$
19: **end for**
20: $f(\cdot) \leftarrow LSTM(\cdot)$

---

## 4. Experiment Results

### 4.1. Experimental Setup

This study proposes a deep learning method for circuit simulation, which involved training a deep learning model using Python and the PyTorch framework in the WSPICE simulator based on SPICE3f5. WSPCIE is an in-house simulator and integrates state-of-the-art DC analysis methods (including CEPTA and DPTA). The final model was obtained by training on 745 samples of 5 circuits. To evaluate the performance of the proposed method, it was compared with the conventional PTA algorithm and adaptive PTA algorithm in terms of the total number of NR iterations required for simulation. All circuits used in the experiments were selected from benchmark [25] and laboratory sources. The experiments were conducted on a Windows workstation equipped with a 1.80 GHz Intel (R) Core (TM) i7-8565U CPU and 512 GB memory. The proposed method demonstrated superior efficiency and robustness in circuit simulation.

### 4.2. Acceleration Simulation Efficiency

To demonstrate the enhanced efficiency of the proposed method, we conducted a comparison with the conventional and adaptive methods. Pseudo-elements were inserted

into the transistors between each node and ground for all three time-step control methods. The diagonal embedding position [19] yielded superior convergence. The efficacy of each method was evaluated by considering the number of NR iterations required. The results, as presented in Table 3, indicate that the proposed method achieves up to 61.32x and 61.13x improvements over the conventional and adaptive methods, respectively, in terms of NR iterations in DPTA. DPTA, which is a variant of PTA, overcomes oscillation problems through artificially enlarged damping effects in the numerical integration algorithm. Furthermore, we provide the circuit characteristics, including nodes, BJT, MOS, etc., in Table 4, to demonstrate the algorithm's superior generalization capability, which is a crucial indicator of deep learning.

**Table 3.** Simulation efficiency for DPTA.

| Circuit | Number of Iters | | | Speedup | |
|---|---|---|---|---|---|
| | Conventional | Adaptive | Ours | vs. Conventional | vs. Adaptive |
| nagle | 2093 | 1948 | 524 | 3.99 | 3.72 |
| ab_ac | 3961 | 3947 | 134 | 29.56 | 29.46 |
| ab_integ | 4540 | 4406 | 219 | 20.73 | 20.12 |
| ab_opamp | 2417 | 2536 | 263 | 9.19 | 9.64 |
| e1480 | 5553 | 5514 | 197 | 28.19 | 27.99 |
| mosrect | 838 | 826 | 52 | 16.12 | 15.88 |
| schmitfast | 5681 | 5691 | 99 | 57.38 | 57.48 |
| slowlatch | 9382 | 9353 | 153 | 61.32 | 61.13 |
| fadd32 | 1968 | 1859 | 137 | 14.36 | 13.57 |
| TADEGLOW6TR | 145 | 102 | 148 | 0.98 | 0.69 |
| THM5 | 5331 | 5324 | 114 | 46.76 | 46.70 |

**Table 4.** Circuit characteristics with devices.

| Circuit | Nodes | eqn | bjt | mos2 | mos3 | c | r | v |
|---|---|---|---|---|---|---|---|---|
| nagle | 26 | 54 | 23 | 0 | 0 | 1 | 11 | 5 |
| ab_ac | 25 | 28 | 0 | 31 | 0 | 22 | 1 | 3 |
| ab_integ | 28 | 32 | 0 | 31 | 0 | 24 | 3 | 4 |
| ab_opamp | 28 | 31 | 0 | 31 | 0 | 24 | 4 | 3 |
| e1480 | 145 | 204 | 0 | 28 | 0 | 17 | 130 | 3 |
| mosrect | 6 | 10 | 0 | 4 | 0 | 0 | 2 | 2 |
| schmitfast | 5 | 19 | 0 | 6 | 0 | 0 | 0 | 2 |
| slowlatch | 12 | 37 | 0 | 0 | 14 | 0 | 1 | 5 |
| fadd32 | 161 | 178 | 0 | 288 | 0 | 25 | 0 | 17 |
| TADEGLOW6TR | 18 | 18 | 0 | 3 | 0 | 0 | 18 | 1 |
| THM5 | 26 | 26 | 9 | 0 | 0 | 0 | 0 | 1 |

Moreover, we have specifically chosen two practical circuits, namely "THM5" and "slowlatch", for detailed analysis. The former comprises 9 BJT transistors and 1 voltage source, and the latter includes 14 MOS transistors and 5 voltage sources. Generally, time-step control in PTA can be divided into two stages, namely, the search phase and the convergence phase. As depicted in Figure 8, the voltage waveforms of node 10 and node 5 in circuit "THM5" and node 15 and node 17 in circuit "slowlatch" show that the solution of each pseudo-step is closely approximating the final solution during the convergence phase. The proposed algorithm is capable of providing a larger time-step, which reduces the number of discrete time points required during the convergence stage. The subfigures in Figure 8a,b demonstrate the voltage waveforms of node 10 and node 5 in circuit "THM5" during the search phase. It is evident that the time step during the search phase is small and the number of time points is large, due to the voltage changing from zero to full value during the search phase, resulting in relatively significant fluctuations in the voltage waveform. The use of a small time step is crucial to ensure the convergence of NR. Similar

observations can be made for the circuit "slowlatch". Additionally, the proposed algorithm ensures the continuity of the voltage waveform and avoids causing oscillations, thereby increasing its practicality.

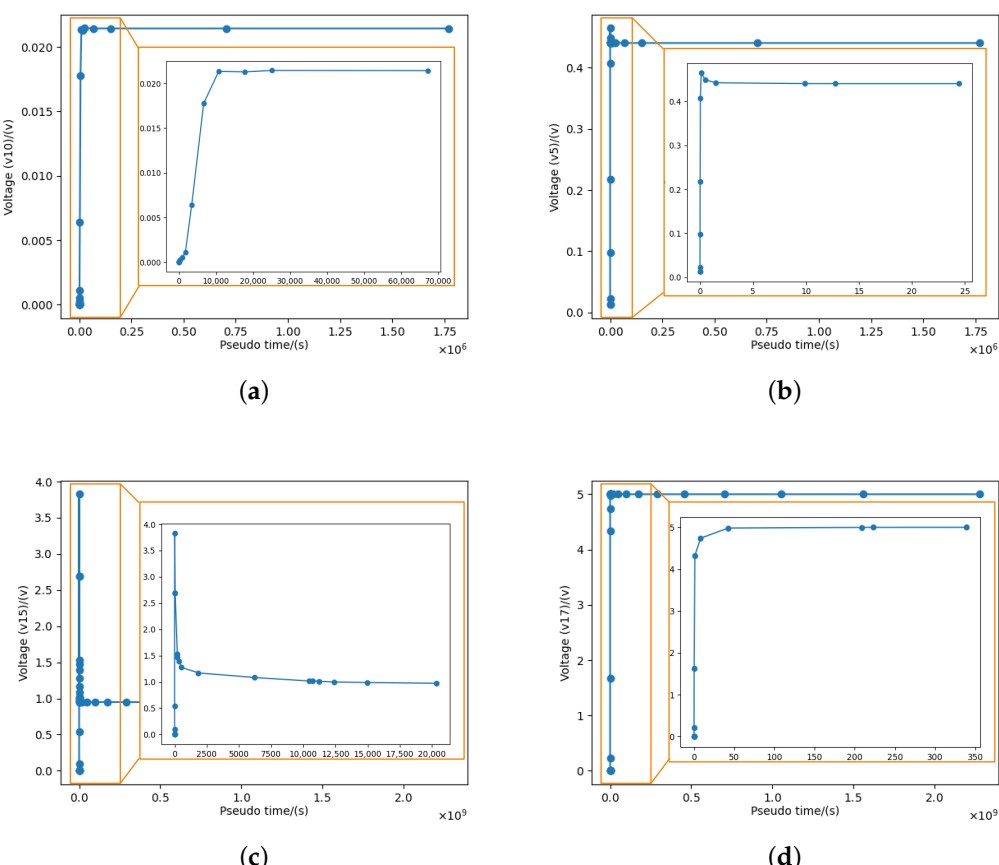

**Figure 8.** The voltage waveforms of "THM5" and "slowlatch" at different nodes under the proposed method. (**a**) The node 10 voltage waveform of the "THM5" circuit. (**b**) The node 5 voltage waveform of the "THM5" circuit. (**c**) The node 15 voltage waveform of the "slowlatch" circuit. (**d**) The node 17 voltage waveform of the "slowlatch" circuit.

*4.3. Improvement Simulation Convergence*

Note that the interesting thing to point out is that for some DPTA non-convergence (N/A) cases shown in Table 5, the proposed method makes them converge. This is particularly useful for PTA based on SPICE because non-convergence issues are extremely difficult for the simulator to deal with, and the cause is often unable to be accurately located. Therefore, the proposed algorithm enhances the robustness of DPTA.

**Table 5.** Improvement convergence for DPTA on some circuits.

| Circuits | Convergence | | |
|---|---|---|---|
| | **Conventional** | **Adaptive** | **Ours** |
| bjtff | N/A | N/A | 144 |
| schmitslow | N/A | N/A | 170 |
| toronto | N/A | N/A | 316 |
| add20 | N/A | N/A | 345 |
| mem_plus | N/A | N/A | 692 |
| ram2k | N/A | N/A | 280 |
| jge | N/A | N/A | 654 |

The proposed algorithm's benefits are exemplified using a sizeable circuit, denoted as "add20", which comprises 958 mos transistors and 42 voltage sources. One portion of the voltage waveform of node 20 in the "add20" circuit under the conventional time-step control approach is depicted in Figure 9a. It is evident that the voltage waveform exhibits oscillations and causes non-convergence, owing to limitations in the conventional time-step control approach. In contrast, the voltage waveform of the same node by using the proposed algorithm, as shown in Figure 9b, mitigates oscillations and enhances the convergence of DPTA.

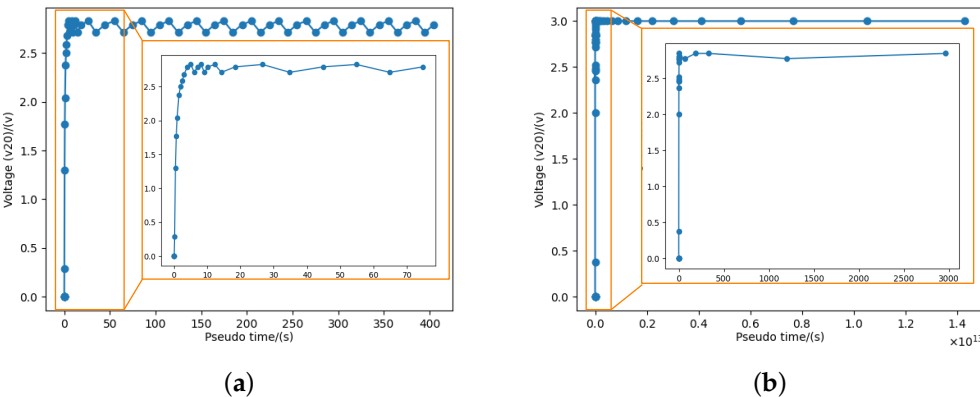

|            (a)            |            (b)            |

**Figure 9.** Comparison of the node voltage waveform on the "add20" circuit under two time-step control methods. (**a**) The voltage waveform of node 20 under conventional time-step control. (**b**) The voltage waveform of node 20 under the proposed method.

### 4.4. Ablation Experiment

To demonstrate the efficacy of feature selection through random forest, we conducted a comparative analysis of the simulation performance of the LSTM memory without the use of random forest for feature selection. To ensure fairness in our assessment, we utilized identical training and test sets for the ablation experiments. The results of these experiments are shown in Table 6. In most test circuits, the feature-selection method based on random forest is significantly better than that of the non-random forest, which is reflected in the proposed algorithm having fewer NR iterations than the algorithm without a random forest, and the total number of PTA iterations and the number of rejected steps were reduced in most circuits. These findings provide empirical evidence to support the practical efficacy of feature selection through random forest.

**Table 6.** Verify the effectiveness of feature selection based on a random forest.

| Circuit | Number of PTA Iters | | Accepted Step | | Rejected Step | | Number of NR Iters | |
|---|---|---|---|---|---|---|---|---|
| | Ours without RF | Ours | Ours without RF | Ours | Ours without RF | Ours | Ours without RF | Ours |
| nagle | 111 | 69 | 91 | 43 | 20 | 26 | 672 | 524 |
| ab_ac | 66 | 39 | 61 | 38 | 5 | 1 | 265 | 134 |
| ab_integ | 85 | 42 | 73 | 37 | 12 | 5 | 402 | 219 |
| ab_opamp | 87 | 52 | 70 | 43 | 17 | 9 | 430 | 263 |
| e1480 | 50 | 27 | 27 | 31 | 23 | 5 | 369 | 197 |
| mosrect | 18 | 14 | 16 | 14 | 2 | 0 | 84 | 52 |
| schmitfast | 50 | 24 | 48 | 24 | 2 | 0 | 176 | 99 |
| slowlatch | 60 | 37 | 56 | 36 | 4 | 1 | 264 | 153 |
| fadd32 | 73 | 33 | 67 | 31 | 6 | 2 | 284 | 137 |
| TADEGLOW6TR | 15 | 26 | 15 | 20 | 0 | 6 | 70 | 148 |
| THM5 | 142 | 29 | 122 | 27 | 20 | 2 | 127 | 114 |

### 5. Conclusions

In this paper, a novel time-step control method enhanced by deep learning was applied to DPTA for completing nonlinear DC analysis efficiently — in particular, a hybrid searching

strategy that is coarse and fine grained for identifying the optimal time step, addressing the key issue that it cannot be theoretically defined. This approach also provides fine-grained labels for the training set. Furthermore, feature selection based on random forest and a two-stage data processing strategy provides a fine dataset for the training model, which improves the training speed and prediction accuracy of model. Finally, the numerical examples demonstrated a fine speedup to 61.32 times compared with other time-step control methods. Fortunately, the proposed method can also improve the non-convergence problem caused by step control in some circuits.

**Author Contributions:** Methodology, Z.J. and X.W.; software, H.P., X.Z. and Z.Y.; original draft preparation, Z.J., X.W., H.P., Z.Y., Y.B. and X.Z.; review and editing, D.N., X.W. and Z.Z.; analysis and validation, X.Z. and Y.B. All authors have read and agreed to the published version of the manuscript.

**Funding:** This work was supported by National Key RD Program of China (Grant No. 2022YFB4400400), and the Key Program of the National Natural Science Foundation of China (Grant Nos. 62234010, 62204265).

**Data Availability Statement:** All the data supporting the reported results have been included in this paper.

**Conflicts of Interest:** The authors declare no conflict of interest.

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
