# Peer review of "Accelerating DC Circuit Simulation through Feature Selection and LSTM-Based Time-Step Control"

_electronics, doi:10.3390/electronics12081927_

Round 1

Reviewer 1 Report

Dear Authors,
I would like to thank you for your achievements in this work. The proposed method assures the convergence in the DC analysis of the electric circuits. The improvement in the speed of the simulation is based on variable time step estimated by neural network model. The work is really interesting for many researchers in the field of circuit simulation. The following points may improve the over all representation:
1) Refer to other numerical method for solving hard nonlinear equations, like Powell's dog leg method and Levenberg–Marquardt.
2) What is the criteria for selecting the training sets of random forest.
3) How can we assure that the trained model of Long Short-Term Memory (LSTM) will be suitable for all types of circuits?
4) In Fig. 5, if NR converge and PTA diverge we may reach the maximum time step, and we will assume it as the optimal time step.
5) In Fig. 5, revise the condition (G > Gmin)
6) Fix the font in Table 2.
7) Why do you select Spice3f5 while there are many alternatives like LTspice and SIMetrix?
8) Try to fix the zoom and the font of figures 8, and 9.
9) In Table 4, try to represent the results relative to the time, not the number of iterations.

Thanks for your time and efforts.
My best regards

Reviewer 2 Report

The paper tackles an important topic, However, several issues need to be addressed in the paper.  some of the comments are presented below:  

  1. The manuscript needs to be revised. I can see many formatting issues. For example  
    •  You need to have a space between the word and the citation packet.  
    •  The terms “LSTM-based Time-step” need to have the first letter of based and step uppercase.  
  2.  All the acronyms need to be defined properly. For example, DC in multiple locations is not defined, don’t presume everyone will know it.  
  3.  Don’t use acronyms for single-used terms. For example, you used  VLSI in the abstract without defining it and it was only used one time in the manuscript.   
  4.  You need to discuss more related works as I can see several works discussing similar topics. see the following reference (DOI: 10.1109/ACCESS.2022.3222269). 
  5.  It is hard to follow the equations, you need to define the symbols clearly. For example, Eqs. (1-12) have no definitions of the symbols used in the equations, except Eq. 8. You need to define each and every term in the equations. Also, it is suggested to cite your equations to provide more authentic derivations.  
  6.  Most of the references are old. can you please justify that? Only references 12 and 19 are recent references. 

Reviewer 3 Report

As the complexity of electrical circuits increases, pseudo-transient analysis (PTA) becomes computationally intensive, primarily due to the time step requirements. This article proposes a new method to determine the optimal time step for the PTA.
The authors present a new method that involves deep learning techniques to discern the distinct time-step necessities, a hybrid searching strategy to find the optimal time-step and a long-short term (LSTM) memory network to process the timing information. Also, random forest is used for feature importance selection to accelerate the training speed and consequently improve the speed and accuracy of the analysis.
A considerable speedup (61.32) can be observed using the proposed method for some test circuits.

The language used for the writing of the paper, and the formatting requires minor revision. Therefore, it may be advisable for the work to be read by someone more familiar with English.

The introduction is relatively short but offers enough basic information so that the reader can get a general idea about the issue that requires to be solved.

The part of the paper that describes the theoretical part of the proposed solution is quite well structured, but the part with the presentation of the practical system that leads to the experimental results is relatively brief.

Presenting the LSTM algorithm on page 11 as a flow chart would allow a better understanding.

Were the experimental results obtained using wspice? As far as I know, this is an online simulation platform.

The paper presents the hardware specification for a PC. What data is processed on that PC and how is it linked to the online simulation platform?

In the Abstract, it is mentioned that the method accelerates the DC analysis, in the best case presented in the experimental results, by 61.13 times. Does this also refer to the analysis's effective duration (time)? And if so, isn't the duration related to the hardware used to run the analysis?

I assume that by using the method described in the paper, part of the data required for the simulation were obtained locally and sent to the online platform, while in the absence of the proposed algorithm, all the data were obtained in the online environment. Does this not affect the data obtained in Table 4?

The Conclusion paragraph is concise, and it should be expanded.

Reviewer 4 Report

This is a very well written paper detailing a method to quickly estimate the "optimal" time step for circuit simulation. The results show a remarkable improvement and the techniques used are well described and convincing.

My only suggestion is to better present the feature used and selected for the LSTM. Table 2 and Figure 6 are a little confusing because features are numbered from 1 to 7 in the table and from 1 to 17 in the figure, but clearly feature #1 in the table is not the same as #1 in the figure. Maybe using matching numbers might help readability.

Last column of Table 5 should probably be "ours" instead of "ous".

Round 2

Reviewer 2 Report

Thank you for replying to the comments and improving the manuscript. However, I am unconvinced about how the manuscript dealt with the references. from the reviewer's point of view, references are as important as other parts of the manuscript. they can indicate how deeply this work is discussing the challenges presented and make sure that the results are not biased.  Therefore, I would rather leave this point to the Editors.
